# The high burden of comorbidities in Aboriginal and Torres Strait Islander Australians living with chronic hepatitis B in Far North Queensland, Australia, and the implications for patient management

Jordan Riddell[1], Allison Hempenstall[2], Yoko Nakata[2], Sandra Gregson[2], Richard Hayes[2], Simon Smith[1], Marlow Coates[2], Lizzie Charlie[2], Christine Perrett[2], Victoria Newie[2], Tomi Newie[2], Sharna Radlof[1], Josh Hanson[1,3]*

**1** Department of Medicine, Cairns Hospital, Cairns, Queensland, Australia, **2** Torres and Cape Hospital and Health Service, Queensland, Australia, **3** Kirby Institute, University of New South Wales, Sydney, Australia

* jhanson@kirby.unsw.edu.au

**Data Availability Statement:** Data cannot be shared publicly because of the ethical protections

## Abstract

### Background

Aboriginal and Torres Strait Islander Australians living with chronic hepatitis B virus (HBV) infection have a significant burden of hepatocellular carcinoma (HCC). The prevalence of comorbidities that increase the risk of HCC in this population is incompletely defined.

### Methods

This cross-sectional study was performed in remote tropical Queensland, Australia in January 2021. All individuals living with chronic HBV in the region were identified; the prevalence of relevant comorbidities was determined by reviewing medical records.

### Results

All 236 individuals in the cohort identified as Aboriginal and Torres Strait Islander Australians; their median (interquartile range (IQR)) age was 48 (40–62) years; 120/236 (50.9%) were female. Of the 194/236 (82.2%) engaged in HBV care, 61 (31.4%) met criteria for HBV therapy and 38 (62.2%) were receiving it. However, 142/236 (60.2%) were obese, 73/236 (30.9%) were current smokers and 57/236 (24.2%) were drinking alcohol hazardously; 70/236 (29.7%) had ≥2 of these additional risk factors for HCC, only 43/236 (18.2%) had none. Among the 19 patients with confirmed cirrhosis, 9 (47%) were obese, 8 (42%) were currently—or had a history of—drinking alcohol hazardously and 5 (26.3%) were current smokers. Patients also had a median (IQR) of 3 (2–4) cardiovascular risk factors (cigarette smoking, hypertension, impaired glucose tolerance, dyslipidaemia, renal impairment/proteinuria). Only 9/236 (3.8%) did not have one of these 5 comorbidities.

of the Queensland Public Health Act of 2005. However, data are available from the Far North Queensland Human Research Ethics Committee (contact via email FNQ_HREC@health.qld.gov.au) for researchers who meet the criteria for access to confidential data.

**Funding:** The authors received no specific funding for this work.

**Competing interests:** The authors have declared that no competing interests exist.

## Conclusions

Aboriginal and Torres Strait Islander Australians living with chronic HBV in this region of remote Australia have a high engagement with HBV care and the majority of individuals eligible for antiviral therapy are receiving it. However, a significant comorbidity burden increases their risk of cirrhosis, HCC, and premature death. It is essential to integrate chronic HBV care with management of these comorbidities—rather than focusing on HBV alone—to achieve optimal health outcomes.

## Introduction

Australia's burden of chronic hepatitis B (CHB) is disproportionately borne by its first people, Aboriginal and Torres Strait Islander Australians (hereafter respectfully referred to, collectively, as Indigenous Australians). The national prevalence of CHB in Indigenous Australians is estimated to be 2.0%, compared to a prevalence of 0.9% in the general population [1]. This contributes to a 4–6 times higher incidence of hepatocellular carcinoma (HCC) in Indigenous Australians, with CHB present in 25.3% of Indigenous Australians diagnosed with HCC compared to 9.9% of non-Indigenous Australians with this cancer [2, 3].

The optimal approach to reducing HCC-related mortality among Indigenous Australians living with CHB remains incompletely defined. The prescription of antiviral therapy, when indicated, is clearly essential—and highly cost-effective—although HCC can still develop on therapy [4, 5]. To identify HCC earlier in those both on—and off—antiviral therapy, it is recommended that all Indigenous Australians ≥50 years of age receive 6-monthly ultrasound surveillance for HCC, with or without alpha-fetoprotein (AFP) [6, 7]. However, it is challenging to provide this HCC surveillance for Indigenous Australians living in remote Australia, indeed, the cost-effectiveness of the strategy has yet to be established [8–11]. Access to imaging is usually limited in these locations and there are many other health conditions—particularly macrovascular disease, diabetes mellitus, chronic lung disease and chronic kidney disease—that compete for finite resources [9, 12].

These other health conditions, strongly linked to socioeconomic disadvantage, have their own attributable morbidity and mortality [12–14]. Furthermore, they also make a significant contribution to the poor outcomes seen in Indigenous Australians diagnosed with HCC. Indeed, in a large Australian study of patients diagnosed with HCC, a higher comorbidity burden and remote residential address were the strongest predictors of HCC-related mortality; Indigenous status was not, in fact, independently prognostic [2].

The current prevalence of CHB in some communities in the Torres Strait Islands (TSI) and Northern Peninsula Area (NPA) in remote tropical Australia is similar to that seen in some Southeast Asian countries [15, 16]. Other comorbidities that increase cirrhosis and HCC risk are also common; most importantly obesity and hazardous alcohol consumption [17–19]. There is also a significant burden of other chronic diseases—particularly macrovascular disease and complications of the metabolic syndrome—that have an even greater heath impact [13, 20]. This study was performed to determine the burden of these comorbidities in an effort to inform the delivery of optimal—and holistic—chronic disease care to the people living with CHB in the region.

## Methods

This cross sectional study was performed in January 2021 in the TSI and NPA region of remote, Far North Queensland (FNQ), Australia (Fig 1). The area covers approximately 48,000

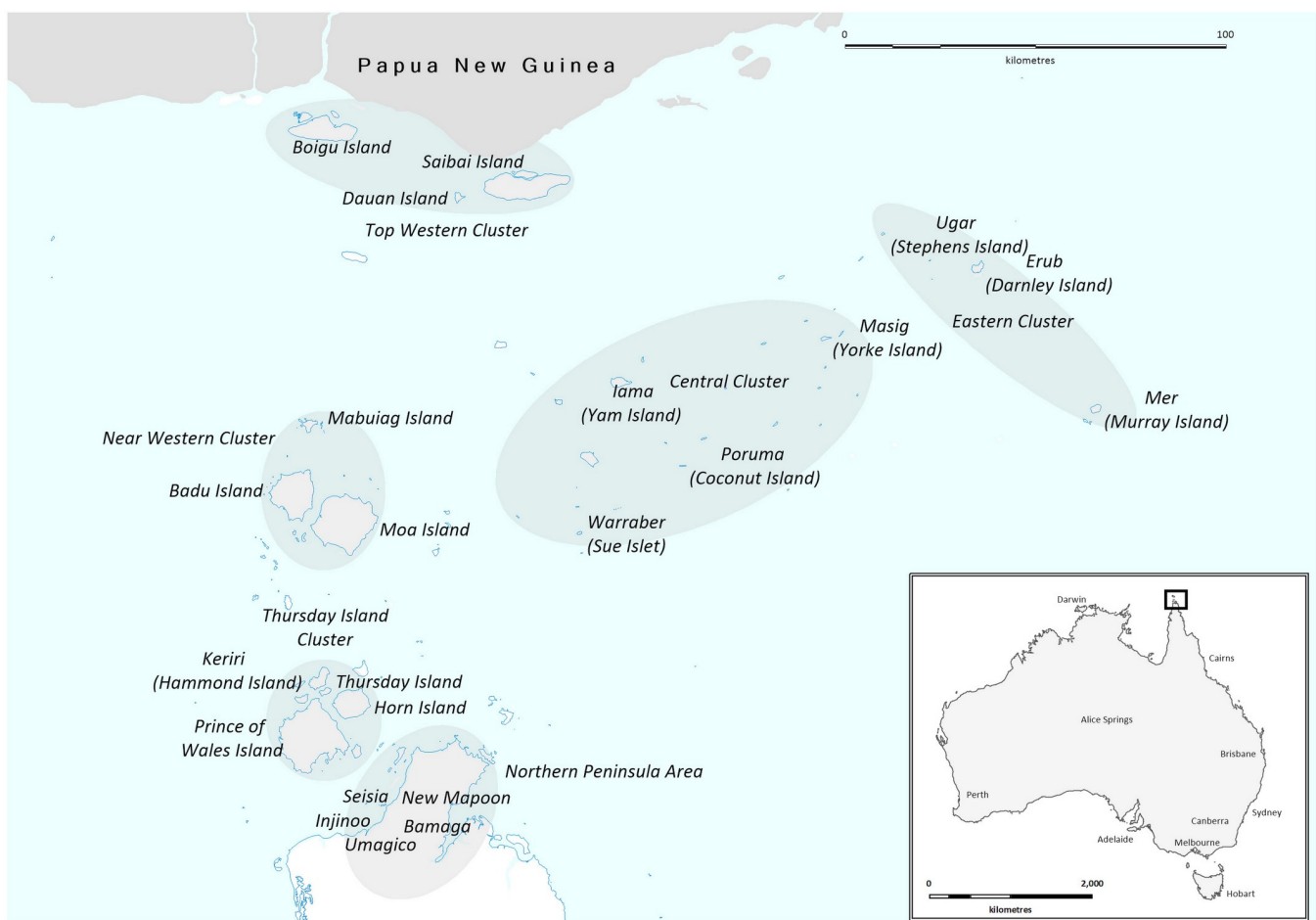

**Fig 1. The study region of the Torres Strait Islands and Northern Peninsula Area in remote Far North Queensland, Australia.** The map was constructed using mapping software (MapInfo version 15.02, Connecticut, USA) using data provided by the State of Queensland (QSpatial). Queensland Place Names—State of Queensland (Department of Natural Resources, Mines and Energy) 2019, available under Creative Commons Attribution 4.0 International licence https://creativecommons.org/licenses/by/4.0/. 'Coastline and state border–Queensland—State of Queensland (Department of Natural Resources, Mines and Energy) 2019, available under Creative Commons Attribution 4.0 International licence https://creativecommons.org/licenses/by/4.0/.

km$^2$ and has an estimated resident population of approximately 11000, over 90% of whom identify as an Indigenous Australian [21]. The population is spread across 18 island communities on the Torres Strait and 5 communities in the NPA (Australian Bureau of Statistics Local Government Areas 36950, 36960 and 35780). People living with CHB that were resident in the region were eligible for inclusion in the study.

Hepatitis B is a notifiable disease in Queensland with all notifications reported to the Queensland Notifiable Conditions System (NOCS) database. This database provides a data extract for a local database (the FNQ HBV database) that allows local clinicians to identify—and expedite the care of—people living with CHB in the region. The FNQ HBV database was used to identify individuals eligible for this study. CHB was defined as ≥2 documented positive hepatitis B surface antigen (HBsAg) serology results ≥6 months apart. Further demographic, clinical, laboratory and radiology data were collected using electronic medical records and laboratory and radiology databases.

All individuals receiving care in Queensland's public health system, are asked whether they identify as an Aboriginal Australian, a Torres Strait Islander Australian, both or neither; this

was recorded. Patients were defined as being engaged in care if they were receiving anti-HBV therapy (tenofovir disoproxil fumarate (TDF) or entecavir, both are provided to local patients free-of-charge by the public health system) or had a quantitative HBV viral load requested in the 2020 calendar year. Patients were said to have cirrhosis if their most recent AST to Platelet Ratio Index (APRI) score was >2 [22], if they had a transient elastography score of >11.7kPa [23], if they had a diagnosis of cirrhosis on imaging that had been reported by a specialist radiologist, or if a diagnosis of cirrhosis was documented in the medical record. While some patients were able to have elastography performed, the machine in use was not able to determine a controlled attenuation parameter (CAP) score; accordingly fatty liver was said to be present if it was described in the report of a specialist radiologist.

Patients were staged using the Australasian Society for HIV, Viral Hepatitis and Sexual Health Medicine (ASHM) guidelines into immune tolerance, immune clearance, immune control or immune escape phases [24]. Patients who had two negative HBsAg tests ≥ 6 months apart were said to have cleared the virus. Those without serology or an HBV viral load in 2020 were classified as "unable to be staged". Patients were deemed to be eligible for treatment if they were in the immune clearance or immune escape stage with evidence of active hepatitis (a serum alanine aminotransferase (ALT) >45 IU/L) or had cirrhosis [25]. If patients were deemed eligible for treatment but were not on therapy, the medical record was reviewed to determine the reason.

The participants' medical records were examined for significant comorbidities. A history of any co-existing liver disease was documented. Obesity was defined as a body mass index (BMI) >30 kg/m$^2$ (using the most recent recorded height and weight). The World Health Organization definition of metabolic syndrome was used to define the features of this syndrome, although in the absence of a recorded waist measurement in almost all the patients, if a patient had a BMI of ≥30 mg/kg$^2$ and two or more of impaired glucose tolerance, hypertension or dyslipidaemia, they were said to satisfy a modified definition of metabolic syndrome [26]. Impaired glucose tolerance was defined as a glycosylated haemoglobin (HbA1c) ≥5.7% at any time [27]. Hypertension was said to be present if a blood pressure of >140/90 was recorded on two occasions or the patient was receiving anti-hypertensive therapy [28]. Dyslipidaemia was said to be present if a patient had a reduced high-density lipoprotein (≤0.9 mmol/L in males, ≤1.0 mmol/L in females) or a serum triglyceride level ≥1.695 mmol/L at any time [29]. The presence of renal impairment (estimated glomerular filtration rate (eGFR) <60 ml/minute/1.73m$^2$) and proteinuria (urinary albumin: creatinine excretion ratio ≥2.5 g/mol in males or ≥3.5 g/mol in females in the absence of concurrent illness) was also sought [30].

Participants' electronic medical records—which are updated at each clinic review—were reviewed to determine if they were a current, past, or never smoker. They were also reviewed to determine if there was current—or a history of—hazardous alcohol use (regular consumption of >10 units of alcohol per week or regular binges of greater than 4 units per day) [31]. Cigarette smoking, impaired glucose tolerance, hypertension, dyslipidaemia, and renal impairment/proteinuria were said to be modifiable cardiac risk factors [32]. A BMI of ≥30 kg/m$^2$, current smoking or current hazardous alcohol use were defined as additional risk factors for HCC [19].

The participants' medications were reviewed. The use of daily aspirin or statin therapy was determined for each patient. The number of regular medications (including ingested, inhaled but not implanted medications) was also recorded. Polypharmacy was defined as the prescription of ≥ 5 medications [33].

Data were de-identified, entered into an electronic database (Microsoft Excel) and analysed using statistical software (Stata version 14.2). Groups were analysed using the Kruskal-Wallis, chi-squared and Fisher's exact test, where appropriate. The study was approved by the Far

North Queensland Human Research Ethics Committee (QCH106-1082). As the data were de-identified, the committee waived the requirement for informed consent.

## Results

Of the 276 individuals identified as potential study participants, 13 (4.7%) had cleared their HBV, 6 (2.2%) had died and 21 (7.6%) had moved from the area, leaving 236 individuals satisfying inclusion criteria for the study; this represented a current local community CHB prevalence of 2.2%. All 236 identified as Indigenous Australians: 217 (92.0%) identified as a Torres Strait Islander, 18 (7.6%) identified as both Torres Strait Islander and Aboriginal and 1 (0.4%) identified as an Aboriginal Australian; 120/236 (50.9%) were female. The median (interquartile range (IQR)) age of the cohort was 48 (40–62) years; 24/236 (10.2%) were born after 1985, the year when vaccination against HBV began in the region.

### Engagement in care and eligibility for treatment

Of the 194/236 (82.2%) engaged in HBV care, 61 (31.4%) met criteria for HBV therapy, 38 (62.2%) of whom were receiving it (19 were receiving TDF and 19 were receiving entecavir). Among the remaining 23, 10 (43.5%) were engaged in case receiving appropriate care (they had been referred for specialist review or were receiving close monitoring while therapy was being discussed). There were 13/61 (21.3%) who were engaged in care and eligible for therapy, but in whom there was no documented plan for therapy; in these patients the opportunity for consideration of treatment appeared to have been missed (Fig 2).

### Ultrasound surveillance

Overall, 128/236 (54.2%) of the patients received a liver ultrasound during the previous 12 months. This included 84/119 (70.6%) of those meeting criteria for HCC screening and 78/104 (75%) of those engaged in care and meeting criteria for HCC screening.

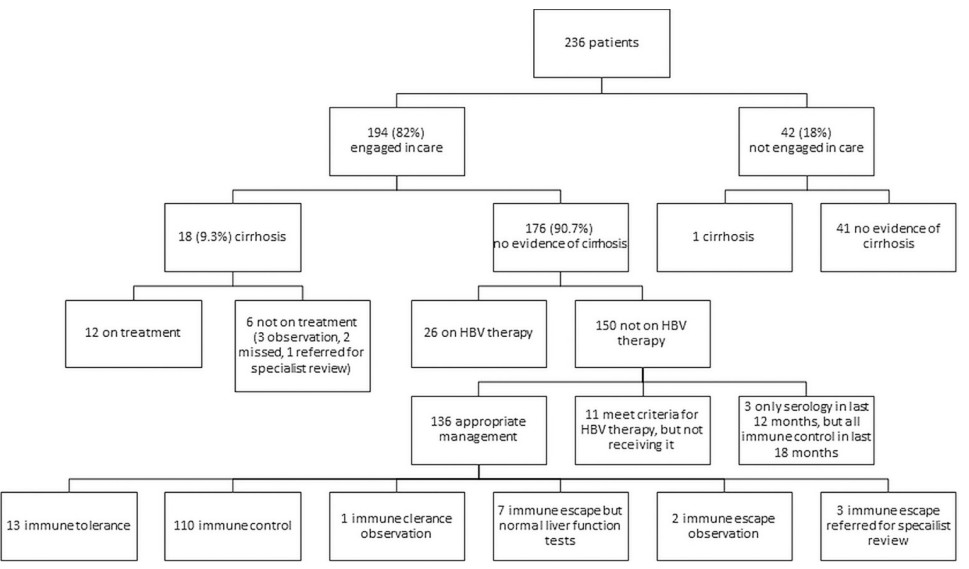

**Fig 2. A flow chart demonstrating the proportion of patients living with chronic hepatitis B in the study region who were engaged in care and who were receiving antiviral therapy (stratified by clinical phenotype).**

## Cirrhosis and HCC

There were 19/236 (8.1%) who met criteria for a diagnosis of cirrhosis; 9/116 (7.8%) males and 10/120 (8.3%) females. Patients with cirrhosis were older than those without cirrhosis (median (IQR): 61 (52–65) compared with 47 (39–60), p = 0.001). Of the 19 patients with cirrhosis, 18 (94.7%) were engaged in care and 12 (66.7%) were on therapy (Fig 2). The single case with an active, confirmed HCC was a 48-year-old, male, overweight former smoker with a history of hazardous alcohol consumption and cirrhosis, who was on anti-HBV therapy. He was diagnosed with multifocal HCC when he presented with ascites; a surveillance ultrasound 8 months prior had not revealed a focal lesion. His AFP at diagnosis (2.7 µg/L) was normal. His case was discussed in a multi-disciplinary team meeting with a transplant centre and as he had decompensated disease, no locoregional or systemic options were available to treat the HCC and a palliative approach was recommended.

## Other liver disease, hazardous alcohol use and cigarette smoking

Among the 217 non-cirrhotic patients, 115 (53.0%) had another liver disease documented, this was fatty liver disease in 112 (97.4%). Two patients had hypoechoic lesions on ultrasound consistent with a haemangioma, the other had a focal area of echogenicity consistent with a haemangioma or fatty change. Current hazardous alcohol use was documented in 57/236 (24.2%), past hazardous use in 28/236 (11.9%). Current tobacco smoking was documented in 73/236 (30.9%), while previous tobacco smoking was recorded in 78/236 (32.6%).

## Obesity and its complications

There were 219/236 (92.8%) who had a height and weight that permitted calculation of a BMI; their median (IQR) BMI was 32.7 (28.7–36.9) kg/m$^2$. There were 198/219 (90.4%) with a BMI >25 kg/m$^2$, 143/219 (65.3%) with a BMI >30 kg/m$^2$, and 82/219 (37.4%) with a BMI >35 kg/m$^2$.

There were 170/236 (72.0%) who had laboratory evidence of impaired glucose tolerance, 106/236 (44.9%) with hypertension, and 190/236 (80.5%) with dyslipidaemia. Of the 219 patients with a recorded BMI, 115 (52.5%) satisfied the modified criteria for the presence of the metabolic syndrome.

## Cardiovascular risk factors

All but 9 (3.8%) patients had at least one modifiable cardiovascular risk factor; 195/236 (82.6%) had ≥2 cardiovascular risk factors (Fig 3). There were 5/236 (2.1%) individuals with a recorded eGFR <30 mL/min/1.73m$^2$ and 31/236 (13.1%) with an eGFR <60 mL/min/1.73m$^2$. There were 94/236 (39.8%) with documented proteinuria.

## Medication burden

Patients were taking a median (IQR) of 2 (0–5) medications; 74 (31.4%) were taking ≥5 medications. There were 76/236 (32.2%) prescribed a statin and 47/236 (19.9%) prescribed regular aspirin.

## Additional risk factors for HCC

In the entire cohort, there were only 43/236 (18.2%) who were not obese, currently smoking or currently drinking alcohol hazardously, while 70/236 (29.7%) had 2 or more of these risk factors for HCC (Fig 4). Among the 19 patients with cirrhosis, only 9/19 (47.4%) were not obese, currently smoking or currently drinking hazardously. Among these 19 patients with cirrhosis,

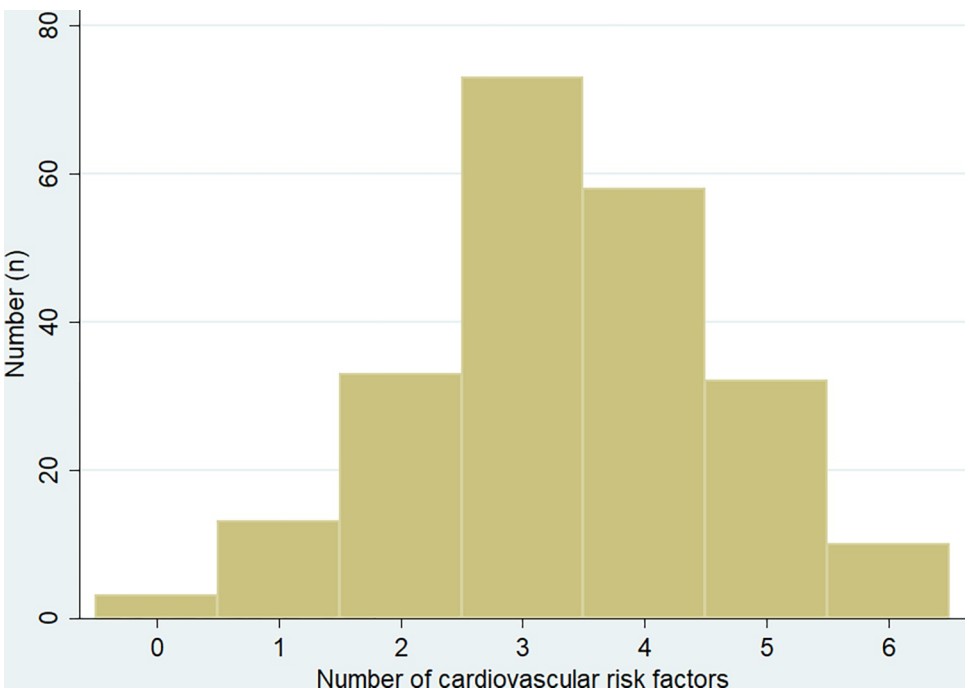

**Fig 3. A histogram showing the number of modifiable risk factors for cardiovascular disease in each of the patients in the cohort.**

9 (47.3%) had a BMI>30 kg/m², 2 (11.1%) were currently drinking alcohol hazardously, while 6 (31.6%) had a history of past hazardous alcohol consumption. Meanwhile, 5/19 (26.3%) were current smokers and 5/19 (26.3%) were past smokers.

## Discussion

Over 80% of individuals living with CHB in this remote region of Australia were engaged in care, one of the highest rates of engagement in Australia, and well above Australia's National Strategy target for engagement of 50% by 2022 [1]. Meanwhile, over 60% of those eligible for therapy were currently receiving antiviral therapy, with almost half of the remaining individuals eligible for therapy having either been referred for specialist review or having elected, in shared decision making with their primary care provider, to defer therapy. However, despite the encouraging rates of engagement and antiviral treatment, over 80% of the individuals in the cohort were obese or were currently smoking tobacco or drinking alcohol in a hazardous manner. If unaddressed, these comorbidities would not only increase the risk of cirrhosis and HCC [34–36], but they would also be expected to have a significant impact on general health outcomes [37–39]. These data emphasise the importance of a more holistic approach to CHB care in the region.

Current Australian guidelines for the management of HBV have a strong emphasis on virological markers of disease, co-infection with other blood borne viruses, indications for antiviral therapy and screening for hepatic complications, particularly HCC [7, 40]. This virological emphasis is understandable as it is currently estimated that 27% of people living in Australia with CHB are undiagnosed, that only 22.6% of are engaged in care, and that only about half of the 20% who are estimated to require treatment are currently receiving it [1]. However there is far less emphasis on identifying and aggressively addressing co-morbidities that can have a significant impact on liver health and associated risk of HCC development [6, 41, 42].

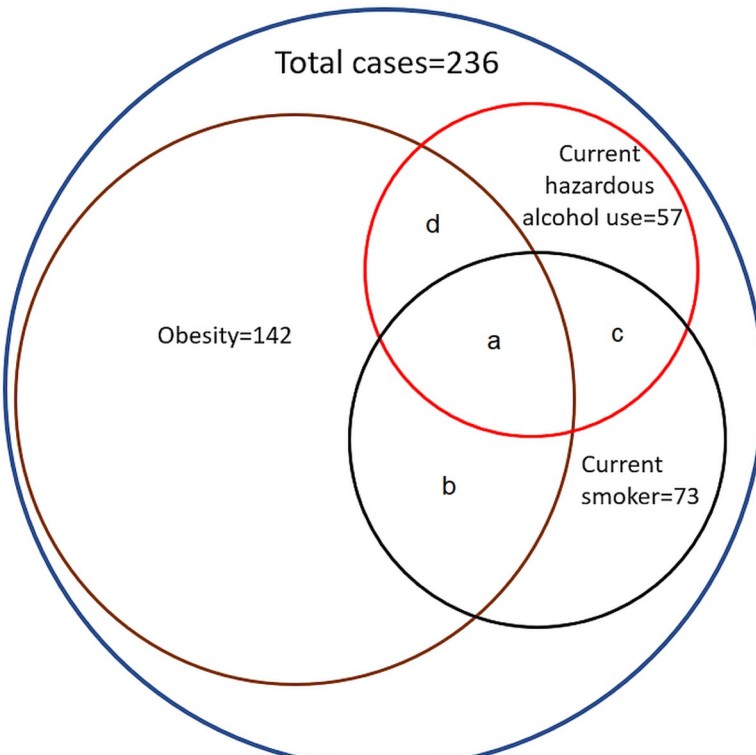

**Fig 4. Venn diagram demonstrating the number of people living with chronic hepatitis B in the region who were also obese, currently smoking or currently drinking hazardously.** [a] 19 individuals were obese, current smokers and were currently drinking alcohol in a hazardous manner. [b] 23 individuals were obese and current smokers. [c] 13 individuals were current smokers and were currently drinking alcohol in a hazardous manner. [d] 12 individuals were obese and were currently drinking alcohol in a hazardous manner.

This is important given the significant impact that these comorbidities—specifically obesity, the metabolic syndrome and smoking and hazardous alcohol use—have on the incidence of cirrhosis and HCC [2, 43–47]. In FNQ, obesity, smoking and hazardous alcohol have an even greater significance given the crucial role that they play in the 5 most common causes of death in the local Indigenous Australians population: coronary heart disease, diabetes mellitus, lung cancer, cerebrovascular disease and chronic obstructive pulmonary disease [48]. There is significant potential overlap between the lifestyle interventions recommended to prevent the development of these conditions, their optimal medical management and the care for people living with CHB [19, 49–53]. An integrated approach which manages CHB and these comorbidities concurrently—rather than the siloed approach that is seen too frequently—would be expected to translate into both better liver and general health outcomes [54, 55]. Although a single individual was diagnosed with HCC in the region in the 2020 calendar year, 25 individuals required aeromedical evacuation from the region for an acute coronary syndrome during the same time period [56].

Other studies to examine CHB-related HCC in Indigenous Australians have provided very limited data about the relative contribution of comorbidities to the development of HCC [16, 57]. A large, multi-jurisdictional study that compared the characteristics and outcomes of HCC in Indigenous and non-Indigenous Australians, identified that comorbidity—quantified using the Charlson Comorbidity Index—was the strongest risk factor for death in the cohort. Indeed, in multivariate analysis of this cohort, Indigenous status was not an independent

predictor of survival [2]. However, comorbidity in this study was determined retrospectively using International Classification of Diseases (ICD) coding of hospital separations, during a period when understanding of fatty liver disease was evolving. The authors acknowledged that it was therefore unlikely to capture precisely the relative contributions of these comorbidities to outcomes [58]. This present cross-sectional study—which uses primary care data—is likely to provide a more accurate assessment of the prevalence of these comorbidities in the community, and therefore the potential role that they play in the development of liver disease.

After appropriate antiviral prescription and optimal management of comorbidities, the incremental value of HCC surveillance in reducing HCC-related mortality in this region is uncertain. Australian national guidelines recommend biannual screening with ultrasound and consideration of AFP testing in people living with CHB who are at increased risk of HCC; this includes all patients with cirrhosis and Indigenous Australians who are older than 50 years [7]. The recommendation for screening in Indigenous Australians is based on a series of 22 HBsAg-positive cases of HCC in the Northern Territory between 2000 and 2011 and an additional 16 cases that were HBcAb-positive, however, the comorbidities of the cases in this series were not presented, which is important as the authors' model suggested that 60% of HCC in Indigenous Australians was due to HBV [57]. Furthermore, the generalisability of the Northern Territory findings to the care of Indigenous Australians living in other parts of remote Australia has not been defined, particularly the contribution of the HBV genotype to clinical outcomes [59]. Northern Territory Investigators have produced an impressive body of work describing the ubiquity and clinical implications of the C4 viral genotype in that jurisdiction [57, 60, 61], however, different HBV genotypes are prevalent outside the Northern Territory, which may have less oncogenic potential [59, 62]. Indeed, is notable that there has not been a single case of HBV-related HCC reported to the Queensland state cancer registry since 2000 in an Aboriginal individual from FNQ, despite a community HBV prevalence of >2% in some Aboriginal communities in the region. Instead, all HBV-related HCC in Australian born residents of FNQ in this period have occurred in Torres Strait Islander Australians [16].

There are other challenges with delivering HCC surveillance to this population. In meta-analyses, the sensitivity of ultrasound for HCC detection ranges from 60–84%, although this falls to 47% for early stage disease [63, 64]. The sensitivity of ultrasound falls even further to 21% in obese (BMI $\geq$30 kg/m$^2$) patients [65]. Transport of ultrasound equipment around the 48,000 km$^2$ region is also challenging and the costs associated with confirmation or exclusion of ultrasound findings are significant as the the nearest computed tomography or magnetic resonance imaging is almost 800km away and accessible only by air. Meanwhile AFP testing offers no additional benefit to ultrasound [66].

However, although over 70% of the cohort that were eligible for HCC surveillance received an ultrasound in the prior 12 months, it was notable that despite the high burden of risk factors for HCC, there was only a single confirmed case of HCC in the cohort. Whilst incomplete screening raises the possibility of underdiagnosis, screening rates (during a period impacted significantly by the global COVID-19 pandemic) were not worse than those reported in Australian metropolitan settings [67].

Indeed, other cancers—many which require more invasive diagnostic testing than is required for HCC diagnosis—are diagnosed more frequently in the region. The incidence of HCC between 2010 and 2018 was 8/100,000/year making it the region's 9$^{th}$ commonest cancer (S1 Table) [68]. It is notable that reducing rates of smoking, obesity and hazardous alcohol consumption would not only tend to reduce the incidence of HCC but would also be expected to have also have a salutary effect on 7 of the 8 more common cancers (lung, breast, colorectal, uterine, cervical, oesophageal and stomach cancer). The financial and logistic investment in attempting to provide biannual ultrasound surveillance may be better directed to

implementing the cancer screening programmes that are unequivocally cost-effective in Australia such as cervix or bowel cancer screening, where specimens—in even remote locations—can be self-collected [69–71]. Alternatively, public health resources could be redeployed in augmenting existing public health strategies that address the underlying carcinogens (such as programmes to assist with smoking cessation, moderating alcohol consumption and reducing weight).

Of course, the difficulties of delivering comprehensive, culturally appropriate, longitudinal chronic disease care and executing public health strategies in a region of almost 50000km$^2$ in remote Australia cannot be ignored [72]. Local CHB management is made more challenging by limited local laboratory and radiology support; there is no local elastography and specialist services are based—with the computed tomography—almost 800km away. The management of other chronic diseases faces similar challenges. Travel between the communities in the region is by ferry or more commonly plane, which is expensive and, in the wet season in this tropical region, sometimes impossible. There is significant staff turnover which results in loss of knowledge and interrupts the delivery of programmatic care. It is therefore notable, that despite these challenges, engagement in CHB care is one of the highest in Australia and the proportion of patients on antiviral therapy, who meet criteria for treatment, is also far higher than is seen in most well-resourced metropolitan centres [1].

This study has several limitations. It is almost certain that some HbsAg-positive individuals living in the region have not been captured in this cohort, however as hepatitis B is a notifiable disease in the state of Queensland, and this register was used to identify patients, it is unlikely that a large number have been missed. Indeed, the prevalence of chronic hepatitis B in the cohort (2.2%) is actually higher than the current estimated national prevalence in Indigenous Australians (2.0%) [1], and any missing patients are unlikely to significantly change the conclusions of the study. Almost 20% of the cohort were not engaged in CHB care, precluding determination of their disease stage and requirement for therapy. Only a few of the patients had received transient elastography, and almost half of the cohort did not receive an ultrasound during the study period, potentially leading to an underestimation of the current burden of cirrhosis, however only 13/234 patients in whom an APRI score could be determined had a score >1.0, suggesting that this is unlikely to be a major issue [22]. Incomplete surveillance—and poor sensitivity of ultrasound testing in obese patients—may have led to an underdiagnosis of HCC in the cohort, however the single case of HCC is similar to the 19 cases of HCC in HbsAg-positive Indigenous Australians in the region between 1999 and 2016 [16] and it is notable that some cancers—including gastric and oesophageal cancer—that require more invasive testing for diagnosis actually have a higher incidence in the region, suggesting that underdiagnosis is unlikely to be a major issue. The study did not include a formal health economic analysis of the local cost-efficacy of the HCC surveillance strategies recommended in national guidelines nor community consultation about the perceived value of this care; both could be examined in future studies. Future research could also examine how best to practically integrate the management of CHB with the care of patients with complex comorbidity living in remote socioeconomically disadvantaged communities in the region.

## Conclusions

This study is one of the first to systematically examine the burden of comorbidities in Indigenous Australians living with CHB and their potential impact on the incidence of HCC. Indeed, with the success of HBV vaccination rollout in Australia, these comorbidities are likely to assume more importance in the future [34, 73]. It emphasises that, even in a region where the engagement in CHB care is amongst the highest reported in Australia, optimal liver outcomes

will not be possible without simultaneously addressing these comorbidities. Optimising the management of comorbidities in Indigenous Australians living in the region with CHB would not only be expected to improve their liver health but also reduce the burden of other chronic diseases that have, potentially, an even greater effect on their long-term health outcomes.

## Supporting information

**S1 Table. The annual incidence of the 9 most common cancers in the study region, the availability of screening programmes and whether tobacco smoking, alcohol use and obesity are risk factors for the disease [68].**
(DOCX)

## Acknowledgments

The authors would like to acknowledge all the health workers who were involved in the care of the patients. The authors would also like to acknowledge the Queensland Cancer Control Analysis Team (QCCAT) who provided data about the incidence of different cancers and the Statistical Services Branch of Queensland Health who provided data about the most common causes of death in the region. The authors would also like to thank Mr Peter Horne for his assistance with the preparation of Fig 1.

## Author Contributions

**Conceptualization:** Josh Hanson.

**Data curation:** Jordan Riddell, Allison Hempenstall, Yoko Nakata, Sandra Gregson, Richard Hayes, Sharna Radlof.

**Formal analysis:** Josh Hanson.

**Investigation:** Jordan Riddell, Allison Hempenstall, Sharna Radlof, Josh Hanson.

**Methodology:** Josh Hanson.

**Project administration:** Allison Hempenstall, Lizzie Charlie, Christine Perrett, Victoria Newie, Tomi Newie, Sharna Radlof, Josh Hanson.

**Supervision:** Simon Smith, Josh Hanson.

**Validation:** Allison Hempenstall, Josh Hanson.

**Writing – original draft:** Jordan Riddell, Josh Hanson.

**Writing – review & editing:** Allison Hempenstall, Yoko Nakata, Sandra Gregson, Richard Hayes, Simon Smith, Marlow Coates, Lizzie Charlie, Christine Perrett, Victoria Newie, Tomi Newie, Sharna Radlof, Josh Hanson.

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
