## [Decision Letter · Decision Letter 0]

25 May 2022

PONE-D-22-10364The high burden of comorbidities in Aboriginal and Torres Strait Islander Australians living with chronic hepatitis B in Far North Queensland, Australia, and the implications for patient managementPLOS ONE

Dear Dr. Josh Hanson,

Thank you for submitting your manuscript to PLOS ONE. After careful consideration, we feel that it has merit but does not fully meet PLOS ONE’s publication criteria as it currently stands. Therefore, we invite you to submit a revised version of the manuscript that addresses the points raised during the review process.

We look forward to receiving your revised manuscript.

Kind regards,

Mohamed Hassany

Academic Editor

PLOS ONE

Journal Requirements:

a) You may seek permission from the original copyright holder of Figure 1 to publish the content specifically under the CC BY 4.0 license.  

Reviewers' comments:

Reviewer's Responses to Questions

**Comments to the Author**

1. Is the manuscript technically sound, and do the data support the conclusions?

Reviewer #1: Yes

2. Has the statistical analysis been performed appropriately and rigorously? 

Reviewer #1: Yes

3. Have the authors made all data underlying the findings in their manuscript fully available?

Reviewer #1: Yes

4. Is the manuscript presented in an intelligible fashion and written in standard English?

Reviewer #1: Yes

5. Review Comments to the Author

Reviewer #1: In this study, Riddell et al present data on the presence of co-morbidities in a cohort of 236 Indigenous Australians living with chronic hepatitis B in Queensland, Australia. They demonstrate that there is a high burden of co-morbid disease that may drive higher rates of advanced fibrosis and hepatocellular carcinoma and use this to argue that management should be more holistic rather than focused exclusively on management of the viral infection.

It is important to note that current Australian guidelines recommend that all Indigenous Australians above the age of 50 undergo HCC surveillance regardless of fibrosis stage or antiviral therapy status. This underlines the relatively high risk of HCC in this group. However, as highlighted by the authors, this recommendation is fraught with logistical difficulties and finite resources are an issue. However, this disproportionate burden needs to be addressed to try to reduce some of the health inequalities that this group of patients experience; with poorer outcomes in HBV-related HCC in this group being attributed to a higher co-morbid burden and poorer socioeconomic status rather than ethnicity in and of itself.

This study has a clear aim and a clear message and by enlarge the data presented is relevant. There are some points that I think would enhance the quality of the manuscript and help the reader.

Other points

1. Do the authors have a sense of how complete screening for HBV is in the geographical area studied? Are the 236 patients self-selecting in that they have better access to healthcare? Having said that the fact that the seroprevalence for HBV was 2.2% suggests that the capture rate was pretty high and this is what is suggested in the discussion.

2. The authors apply a lot of effort into classifying the patients into different HBV disease phases. Whilst this is appreciated, it does not make a significant difference to the central message of the manuscript and the level of detail in this section could perhaps be reduced.

3. Figure 4. It would be helpful for the reader to know what the absolute numbers in the overlap areas are. How many patients had all 3 reported risk factors?

4. How do the authors account for the fact that some patients with cirrhosis were not on antiviral therapy despite the need for this being absolutely clearly defined in all the international guidelines?

5. Although not essential, it would add to the manuscript to know which antivirals the patients were prescribed and if possible the length that they had been on the agents for. Also, it would help to contextualise matters to know if antiviral medication is fully provided by the State or if the patient has any costs to bear.

6. How complete was the data on smoking patterns and alcohol consumption? Was it all contemporaneous?

7. The authors should comment on whether the COVID-19 pandemic had any impact on their findings – e.g the number of patients who had one USS in 2020 compared to the fact that they would have been expected to have had a couple if on HCC surveillance.

8. The authors discuss whether or not HCC surveillance in the studied population is cost effective compared to perhaps investing in bowel cancer or cervical screening. I do not think that their data provides sufficient grounds to justify this statement and this section should be tempered.

6. PLOS authors have the option to publish the peer review history of their article (what does this mean?). If published, this will include your full peer review and any attached files.

Reviewer #1: **Yes: **Ahmed M Elsharkawy

---

## [Author Response · Author response to Decision Letter 0]

8 Jun 2022

Response to reviewers

We thank the Editorial staff and the reviewer, Dr Ahmed M Elsharkawy, for the time that they have taken to review our manuscript and the very helpful comments that they have made to improve the work. Please find below our point-by-point responses to their comments. 

Editorial staff comments

Response: We have reviewed the above templates and believe that our manuscript meets PLOS ONE’s style requirements. Please advise us if there are additional specific changes that we need to make.

Response: Unfortunately, we are unable to make our dataset publicly available. This is because the data are obtained from the Queensland State Notifiable Conditions System (NOCS) and data at an individual level are not available publicly because of the Queensland Public Health Act of 2005. However, data are available from the Far North Queensland Human Research Ethics Committee (contact via email FNQ_HREC@health.qld.gov.au) for researchers who meet the criteria for access to confidential data. 

Our data Availability Statement might read “Data cannot be shared publicly because of the Queensland Public Health Act of 2005. However, data are available from the Far North Queensland Human Research Ethics Committee (contact via email FNQ_HREC@health.qld.gov.au) for researchers who meet the criteria for access to confidential data.”

a) You may seek permission from the original copyright holder of Figure 1 to publish the content specifically under the CC BY 4.0 license. 

Response: We have attached the Content Permission Form which is signed by the Manager of Data and Systems Support at the Queensland State Government’s Department of Resources. Please let us know if you have any additional concerns. We have also added text below Figure 1 to highlight the origin of data used to construct the map and the software that was used to do it. 

“The map was constructed using mapping software (MapInfo version 15.02, Connecticut, USA) using data provided by the State of Queensland (QSpatial). Queensland Place Names—State of Queensland (Department of Natural Resources, Mines and Energy) 2019, available under Creative Commons Attribution 4.0 International licence https://creativecommons.org/licenses/by/4.0/. ‘Coastline and state border–Queensland—State of Queensland (Department of Natural Resources, Mines and Energy) 2019, available under Creative Commons Attribution 4.0 International licence https://creativecommons.org/licenses/by/4.0/.”

Reviewers' comments:

Reviewer's Responses to Questions

Comments to the Author

1. Is the manuscript technically sound, and do the data support the conclusions?

Reviewer #1: Yes

Response: We thank Dr Elsharkawy for reviewing the manuscript and are pleased to hear that he believes that the manuscript is technically sound.

2. Has the statistical analysis been performed appropriately and rigorously?

Reviewer #1: Yes

Response: We are happy that Dr Elsharkawy is satisfied with the statistical analysis.

3. Have the authors made all data underlying the findings in their manuscript fully available?

Reviewer #1: Yes

Response: As noted above, data cannot be shared publicly because of the Queensland Public Health Act 2005. Data are available from the Far North Queensland Human Research Ethics Committee (contact via email FNQ_HREC@health.qld.gov.au) for researchers who meet the criteria for access to confidential data. This is because the data are obtained from the Queensland State Notifiable Conditions System (NOCS) and data at an individual level are not available publicly.

4. Is the manuscript presented in an intelligible fashion and written in standard English?

Reviewer #1: Yes

Response: We are pleased that Dr Elsharkawy is satisfied with the written presentation of the manuscript.

5. Review Comments to the Author

Reviewer #1: In this study, Riddell et al present data on the presence of co-morbidities in a cohort of 236 Indigenous Australians living with chronic hepatitis B in Queensland, Australia. They demonstrate that there is a high burden of co-morbid disease that may drive higher rates of advanced fibrosis and hepatocellular carcinoma and use this to argue that management should be more holistic rather than focused exclusively on management of the viral infection.

It is important to note that current Australian guidelines recommend that all Indigenous Australians above the age of 50 undergo HCC surveillance regardless of fibrosis stage or antiviral therapy status. This underlines the relatively high risk of HCC in this group. However, as highlighted by the authors, this recommendation is fraught with logistical difficulties and finite resources are an issue. However, this disproportionate burden needs to be addressed to try to reduce some of the health inequalities that this group of patients experience; with poorer outcomes in HBV-related HCC in this group being attributed to a higher co-morbid burden and poorer socioeconomic status rather than ethnicity in and of itself.

This study has a clear aim and a clear message and by enlarge the data presented is relevant. There are some points that I think would enhance the quality of the manuscript and help the reader.

Response: We thank Dr Elsharkawy again for reviewing the manuscript. We address each of his specific points below.

Other points

1. Do the authors have a sense of how complete screening for HBV is in the geographical area studied? Are the 236 patients self-selecting in that they have better access to healthcare? Having said that the fact that the seroprevalence for HBV was 2.2% suggests that the capture rate was pretty high and this is what is suggested in the discussion.

Response: Dr Elsharkawy raises an excellent point. It is certainly likely that some cases of chronic hepatitis B in the region have been missed. However, as hepatitis B is a notifiable disease in the state of Queensland, if a patient records a positive HbsAg or HBV DNA test in any laboratory in the state (public or private), it is automatically reported to the state register. As the state register of notified cases was used to identify cases for this study, we are unlikely to have missed many cases as the region is home to communities that have regular health screening, including HBsAg testing. Indeed, the community prevalence of 2.2% reported in the paper is actually higher than the current estimate of prevalence in Aboriginal and Torres Strait Islander Australians (2.0% reported in reference 1) that is determined using census data. 

Even if some cases of chronic hepatitis B have been missed, this is likely to be a relatively small number, and would not be expected to change the main findings of the paper (the high prevalence of comorbidities in the local HbsAg-positive population that increases their risk of liver disease and poor health outcomes generally). We have added some text to the limitations section to address Dr Elsharkawy’s concerns however (lines 376-382).

2. The authors apply a lot of effort into classifying the patients into different HBV disease phases. Whilst this is appreciated, it does not make a significant difference to the central message of the manuscript and the level of detail in this section could perhaps be reduced.

Response: We thank Dr Elsharkawy for raising this issue. The staging of disease into different disease phases allows clinicians to determine whether patients are eligible for anti-HBV therapy. Non-cirrhotic patients in the immune tolerant and immune control phases of their chronic hepatitis B infection are not eligible for therapy under current Australian guidelines. Conversely cirrhotic patients, those in the immune escape phase and those in the immune clearance phase for longer than 6 months should be offered therapy if no contraindication exists. 

It is estimated that presently only about 53% of Australian patients who are eligible to receive anti-HBV therapy are receiving it (reference 1). In our cohort 38/61 (62%) of those eligible to receive therapy were receiving this medication. An additional 10 patients had been referred for specialist review to commence therapy or had, in shared decision making with their primary provider elected to withhold therapy. In only 13/61 (21%) did the opportunity to discuss an inhiation of therapy appear to have been missed (figure 2 and lines 271-274). 

We highlight this point because this shows that the “hepatitis B care” that these patients are receiving is amongst the best in the country. Engagement in care was much higher than the national average (82% versus 23%) and almost 80% of patients eligible for therapy had had treatment discussed with them (versus a national figure of 53%). However, progress in reducing harder endpoints (the incidence of cirrhosis, liver cancer, liver-related mortality, and all-cause mortality) - which after all is the main point of HBV care - will be limited unless other comorbidities are also addressed concurrently (lines 269-280). 

3. Figure 4. It would be helpful for the reader to know what the absolute numbers in the overlap areas are. How many patients had all 3 reported risk factors?

Response: We agree with Dr Elsharkawy, this is an excellent point. We have amended figure 4 accordingly. There were 19 (8%) individuals in the cohort who had all three risk factors.

4. How do the authors account for the fact that some patients with cirrhosis were not on antiviral therapy despite the need for this being absolutely clearly defined in all the international guidelines?

Response: We thank Dr Elsharkawy for highlighting this point. As we describe in figure 2, there were 7/19 patients with known cirrhosis who were not on therapy: 3 were under observation (the risk of flare and decompensation in the setting of poor adherence felt by the clinicians to outweigh the undoubted benefit of this therapy), 1 was not engaged in care, 1 had been referred for specialist review (the primary care provider did not feel confident to start the anti-HBV therapy) and 2 had been apparently missed. 

5. Although not essential, it would add to the manuscript to know which antivirals the patients were prescribed and if possible the length that they had been on the agents for. Also, it would help to contextualise matters to know if antiviral medication is fully provided by the State or if the patient has any costs to bear.

Response: We thank Dr Elsharkawy for highlighting this point. There were 19 patients receiving tenofovir and 19 patients receiving entecavir. The cost of this medication is fully borne by the State. We have amended the text to highlight this point (lines 133-134 and lines 195-196). 

6. How complete was the data on smoking patterns and alcohol consumption? Was it all contemporaneous?

Response: We thank Dr Elsharkawy for highlighting this point. The data on smoking and hazardously alcohol consumption was taken from the patient’s electronic medical record, which is a live document. We have amended the methods to reflect this. These records were reviewed to determine if the patients were a current, past, or never smoker. They were also reviewed to determine if there was a history of - or current - hazardous alcohol use (regular consumption of >10 units of alcohol per week or regular binges of greater than 4 units per day) (lines 166-169).

7. The authors should comment on whether the COVID-19 pandemic had any impact on their findings – e.g the number of patients who had one USS in 2020 compared to the fact that they would have been expected to have had a couple if on HCC surveillance.

Response: We thank Dr Elsharkawy for highlighting this point. The COVID-19 pandemic did indeed impact on the delivery of care during 2020. We have added text in the discussion to highlight this point (lines 348-349).

8. The authors discuss whether or not HCC surveillance in the studied population is cost effective compared to perhaps investing in bowel cancer or cervical screening. I do not think that their data provides sufficient grounds to justify this statement and this section should be tempered.

Response: We thank Dr Elsharkawy for highlighting this point, although we respectfully disagree on this issue. There is significant debate about the cost efficacy of HCC surveillance in metropolitan centres, let alone remote settings where this study was performed. 

Approximately 90% of HCC in HbsAg positive patients occur in patients with cirrhosis, many of whom are ineligible for curative therapy. In Australia only 14.5% of non-Indigenous patients and 6.6% of Indigenous patients with HCC are eligible for curative surgery (reference 2). 

Providing expert ultrasound services to a population of 236 patients spread over a geographical area of 48,000km2, 65% of whom are obese (BMI>30 kg/m2) is challenging. Ultrasound surveillance has a sensitivity for detecting HCC at any stage is 78% and 45% if it is at an early stage. However, the sensitivity falls significantly in obese patients 21% in patients with a BMI ≥ 30 kg/m2 versus 77% in patients with a BMI <30 kg/m2 (https://www.ncbi.nlm.nih.gov/pmc/articles/PMC6940490/).

The costs of the HCC screening programme would also need to consider the fact that the nearest CT scanner (for confirmation or exclusion of HCC in a patient with a lesion on ultrasound) is approximately 800 kilometres away, accessible only by commercial flight. 

In contrast, bowel cancer screening and cervical cancer screening (both of which are more common that liver cancer in the region), can be performed at home, are acceptable to patients, are unaffected by BMI and have been shown to be highly cost-effective https://pubmed.ncbi.nlm.nih.gov/34499374/
https://pubmed.ncbi.nlm.nih.gov/21401458/

We have expanded the discussion to provide more justification for our stance, which we acknowledge needed more supporting data (lines 337-363).

---

## [Decision Letter · Decision Letter 1]

25 Jan 2023

PONE-D-22-10364R1The high burden of comorbidities in Aboriginal and Torres Strait Islander Australians living with chronic hepatitis B in Far North Queensland, Australia, and the implications for patient managementPLOS ONE

Dear Dr. Hanson,

Thank you for submitting your manuscript to PLOS ONE. After careful consideration, we feel that it has merit but does not fully meet PLOS ONE’s publication criteria as it currently stands. Therefore, we invite you to submit a revised version of the manuscript that addresses the points raised during the review process.

We look forward to receiving your revised manuscript.

Kind regards,

Mohamed Hassany

Academic Editor

PLOS ONE

Journal Requirements:

Reviewers' comments:

Reviewer's Responses to Questions

**Comments to the Author**

1. If the authors have adequately addressed your comments raised in a previous round of review and you feel that this manuscript is now acceptable for publication, you may indicate that here to bypass the “Comments to the Author” section, enter your conflict of interest statement in the “Confidential to Editor” section, and submit your "Accept" recommendation.

Reviewer #2: (No Response)

Reviewer #3: All comments have been addressed

Reviewer #4: All comments have been addressed

2. Is the manuscript technically sound, and do the data support the conclusions?

Reviewer #2: No

Reviewer #3: Yes

Reviewer #4: Yes

3. Has the statistical analysis been performed appropriately and rigorously? 

Reviewer #2: No

Reviewer #3: Yes

Reviewer #4: Yes

4. Have the authors made all data underlying the findings in their manuscript fully available?

Reviewer #2: Yes

Reviewer #3: Yes

Reviewer #4: No

5. Is the manuscript presented in an intelligible fashion and written in standard English?

Reviewer #2: No

Reviewer #3: Yes

Reviewer #4: Yes

6. Review Comments to the Author

Reviewer #2: The data gathered and the descriptive analysis done do not support the conclusion made by the authors.

Reviewer #3: Summary of the research and overall impression

The manuscript discusses the high burden of co-morbidities among the indigenous Australian population and the implication for delivery of care to people living with chronic Hepatitis B. It is hoped that with holistic care, the mortality associated with Hepatocellular carcinoma in this group of people and burden of other associated diseases would be reduced. The paper is well written with a clear aim and sound arguments.

The following points should however be addressed.

Minor areas for improvement

1. The whole discussion on HCC surveillance and its cost-effectiveness, though important, does not appear to emanate from the data provided from the study and indeed is not part of the aims and objectives of the study. I recommend just a brief mention of it. It may appear as part of recommendations for further studies.

2. A few grammatical corrections are needed. For instance, the word “had” should be omitted from line 304.

Reviewer #4: Thanks for the modifications done, it is really an important issue discusssing the impact of the comorbidities on HBV and this will have a great impact on the management and follow up of patients

7. PLOS authors have the option to publish the peer review history of their article (what does this mean?). If published, this will include your full peer review and any attached files.

Reviewer #2: **Yes: **Dr. Mohamed Alboraie

Reviewer #3: No

Reviewer #4: No

---

## [Author Response · Author response to Decision Letter 1]

1 Feb 2023

Response to reviewers

We thank the Editorial staff and the reviewers for the time that they have taken to review our manuscript and the very helpful comments that they have made to improve the work. Please find below our point-by-point responses to their comments.

Journal Requirements:

Response: We have reviewed the reference list and believe that it is complete and correct. We have not, to our knowledge, cited any papers that have been retracted. We have not made any changes to the reference list.

Reviewers' comments:

Reviewer's Responses to Questions

Comments to the Author

1. If the authors have adequately addressed your comments raised in a previous round of review and you feel that this manuscript is now acceptable for publication, you may indicate that here to bypass the “Comments to the Author” section, enter your conflict of interest statement in the “Confidential to Editor” section, and submit your "Accept" recommendation.

Reviewer #2: (No Response)

Reviewer #3: All comments have been addressed

Reviewer #4: All comments have been addressed

Response: We thank the reviewers for the time that they have taken to review the prior versions of the manuscript and our replies to the concerns of Reviewer 1 (Dr Ahmed M Elsharkawy). We are pleased to hear that they feel that all his comments have been addressed.

2. Is the manuscript technically sound, and do the data support the conclusions?

Reviewer #2: No

Reviewer #3: Yes

Reviewer #4: Yes

Response: We thank Reviewer 3 and 4 for their positive feedback. However, we are disappointed to hear that Reviewer #2 (Dr. Mohamed Alboraie) does not feel that our work is technically sound. If Dr Alboraie is able to detail any specific concerns that he has, we would be happy to address them. 

3. Has the statistical analysis been performed appropriately and rigorously?

Reviewer #2: No

Reviewer #3: Yes

Reviewer #4: Yes

Response: We thank Reviewer 3 and 4 for their positive feedback. However, we note that Dr. Alboraie has concerns about our statistical analysis. If Dr Alboraie is able to highlight specific deficiencies, we would be happy to try and remedy them.

4. Have the authors made all data underlying the findings in their manuscript fully available?

Reviewer #2: Yes

Reviewer #3: Yes

Reviewer #4: No

Response: Unfortunately, the Queensland Public Health Act of 2005 prevents us from including the full dataset in our submission. The same legislation precludes us depositing the dataset in a public repository. However, interested readers can access the complete deidentified dataset by contacting the Far North Queensland Human Research Ethics Committee. We have provided a statement to this effect in our submission: “Data cannot be shared publicly because of the ethical protections of the Queensland Public Health Act of 2005. However, data are available from the Far North Queensland Human Research Ethics Committee (contact via email FNQ_HREC@health.qld.gov.au) for researchers who meet the criteria for access to confidential data”.

5. Is the manuscript presented in an intelligible fashion and written in standard English?

Reviewer #2: No

Reviewer #3: Yes

Reviewer #4: Yes

Response: We are heartened that Reviewers 3 and 4 both feel that the language is clear, correct, and unambiguous. Unfortunately, Dr Alboraie does not agree. We would be happy to address any specific concerns that he has if his able to highlight them for us.

6. Review Comments to the Author

Reviewer #2: The data gathered and the descriptive analysis done do not support the conclusion made by the authors.

Response: We are disappointed to hear that Dr Alboraie does not feel that the data gathered do not support the conclusion that we have drawn. 

However, we would contend that we have been able to present data that supports our conclusions. 

Our conclusions ( presented in the conclusions section of the abstract) are:

1. Aboriginal and Torres Strait Islander Australians living with chronic HBV in this region of remote Australia have a high engagement with HBV care. 

We would contend that the cohort’s engagement in care was high. The most recent analysis of Australia’s chronic hepatitis B cascade of care suggested that only 22.6% of Australians living with chronic hepatitis B are engaged in care (https://hepatitissa.asn.au/blog/hepatitis-b-targets-australia-23-years-behind-on-care-24-on-treatment/ ) The fact that 194/236 (82.2%) of the cohort were engaged in care (almost four times greater than Australia’s national figure and greater than the country’s 2030 target of 50%) suggests that Aboriginal and Torres Strait Islander Australians living with chronic HBV in this region of remote Australia do indeed have a high engagement with HBV care. 

2. The majority of individuals eligible for antiviral therapy are receiving it. 

61 (31.4%) met national criteria for HBV therapy and 38 (62.2%) - a majority - were receiving it.

3. A significant comorbidity burden increases their risk of cirrhosis, HCC, and premature death. 

We would contend that the fact that patients in the cohort had a median (IQR) of 3 (2-4) cardiovascular risk factors, that 142/236 (60.2%) were obese, that 73/236 (30.9%) were current smokers and that 57/236 (24.2%) were drinking alcohol hazardously does represent a significant comorbidity burden. It is hard to argue that this would not be expected to increase the risk of cirrhosis, HCC, and premature death (liver-related and non-liver related) death.

4. It is essential to integrate chronic HBV care with management of these comorbidities - rather than focusing on HBV alone - to achieve optimal health outcomes.

We would argue that this is self-evident. If we don’t also address the patients’ obesity, hazardous alcohol consumption and tobacco use, we will inevitable not achieve optimal health outcomes, no matter how well we manage their hepatitis B.

Reviewer #3: Summary of the research and overall impression

The manuscript discusses the high burden of co-morbidities among the indigenous Australian population and the implication for delivery of care to people living with chronic Hepatitis B. It is hoped that with holistic care, the mortality associated with Hepatocellular carcinoma in this group of people and burden of other associated diseases would be reduced. The paper is well written with a clear aim and sound arguments.

The following points should however be addressed.

Minor areas for improvement

1. The whole discussion on HCC surveillance and its cost-effectiveness, though important, does not appear to emanate from the data provided from the study and indeed is not part of the aims and objectives of the study. I recommend just a brief mention of it. It may appear as part of recommendations for further studies.

2. A few grammatical corrections are needed. For instance, the word “had” should be omitted from line 304.

Response: We thank reviewer 3 for the time that they have taken to review our paper and their constructive feedback. We acknowledge that we have not performed a health economic analysis of the currently recommended HCC surveillance strategies in the region and therefore we need to be cautious in our discussion around cost-effectiveness of different approaches. We have therefore added this as a limitation of our study and its potential as a subject for detailed future study (lines 393-396). 

However, we would argue that highlighting the burden of comorbidities among people living with chronic hepatitis B in the region is an important focus of the study. It is critical to define the prevalence of comorbidities and their potential impact on HCC incidence as, only then, after the optimisation of their management are we able to determine the incremental value of HCC surveillance. 

We do highlight these points in the introduction where we outline the aims of the study. We note that “The optimal approach to reducing HCC-related mortality among Indigenous Australians living with CHB remains incompletely defined” (lines 76-77). We highlight uncertainty about the cost-effectiveness of HCC surveillance in remote Australia (lines 81-83). We also emphasise the importance of optimising the management of comorbidities particularly hazardous alcohol use and obesity (lines 95-96). We then go on to say, “This study was performed to determine the burden of these comorbidities in an effort to inform the delivery of optimal - and holistic - chronic disease care to the people living with CHB in the region” (lines 99-101), which clearly involves the utility – or lack thereof – of HCC surveillance.

We would contend that having identified a very high rate of comorbidity (cardiovascular risk factors and smoking, obesity and hazardous alcohol use) which all had an impact on the 5 most common causes of death in the local Indigenous Australian population and relatively low rates of HCC (only the 9th most common cancer in the region by incidence), we are justified in suggesting that biannual ultrasound surveillance for HCC (across a geographical area of 48,000 km2) is unlikely to be a very high value investigation for reducing HCC-related mortality. The region reports approximately 1 case of HCC every year (reference 16). In the year of this study, HCC surveillance did not identify a single HCC in the population. 

Indeed, the single case in the cohort with an active, confirmed HCC was diagnosed with multifocal HCC when he presented with ascites; a surveillance ultrasound 8 months prior had been reported as normal. His AFP at diagnosis (2.7 µg/L) was normal. His case was discussed in a multi-disciplinary team meeting with a transplant centre and as he had decompensated disease, no locoregional or systemic options were available to treat the HCC and a palliative approach was recommended (lines 215-222). Although he was two months overdue for his biannual ultrasound, it is unlikely that it would have identified a curable lesion. 

We have corrected the typographical error in line 304, omitting “had” from the line. We would be happy to address any other grammatical errors that the Reviewer or Editorial staff highlight.

Reviewer #4: Thanks for the modifications done, it is really an important issue discussing the impact of the comorbidities on HBV and this will have a great impact on the management and follow up of patients.

Response: We thank Reviewer 4 again for the time that they have taken to review our paper and for their very positive feedback.

---

## [Decision Letter · Decision Letter 2]

27 Mar 2023

The high burden of comorbidities in Aboriginal and Torres Strait Islander Australians living with chronic hepatitis B in Far North Queensland, Australia, and the implications for patient management

PONE-D-22-10364R2

Dear Dr. Hanson

We’re pleased to inform you that your manuscript has been judged scientifically suitable for publication and will be formally accepted for publication once it meets all outstanding technical requirements.

Kind regards,

Mohamed El-Kassas

Academic Editor

PLOS ONE

Additional Editor Comments (optional):

Thanks for the authors for clarifying and responding to reviewers's comments.

Reviewers' comments:

Reviewer's Responses to Questions

**Comments to the Author**

1. If the authors have adequately addressed your comments raised in a previous round of review and you feel that this manuscript is now acceptable for publication, you may indicate that here to bypass the “Comments to the Author” section, enter your conflict of interest statement in the “Confidential to Editor” section, and submit your "Accept" recommendation.

Reviewer #3: All comments have been addressed

2. Is the manuscript technically sound, and do the data support the conclusions?

Reviewer #3: Yes

3. Has the statistical analysis been performed appropriately and rigorously? 

Reviewer #3: Yes

4. Have the authors made all data underlying the findings in their manuscript fully available?

Reviewer #3: Yes

5. Is the manuscript presented in an intelligible fashion and written in standard English?

Reviewer #3: Yes

6. Review Comments to the Author

Reviewer #3: (No Response)

7. PLOS authors have the option to publish the peer review history of their article (what does this mean?). If published, this will include your full peer review and any attached files.

Reviewer #3: No

---

## [Editor Report · Acceptance letter]

29 Mar 2023

PONE-D-22-10364R2 

The high burden of comorbidities in Aboriginal and Torres Strait Islander Australians living with chronic hepatitis B in Far North Queensland, Australia, and the implications for patient management. 

Dear Dr. Hanson:

I'm pleased to inform you that your manuscript has been deemed suitable for publication in PLOS ONE. Congratulations! Your manuscript is now with our production department. 

Kind regards, 

on behalf of

Professor Mohamed El-Kassas 

Academic Editor

PLOS ONE